# Diffusive and arrested transport of atoms under tailored disorder

Fangzhao Alex An[1], Eric J. Meier [1] & Bryce Gadway [1]

Ultracold atoms in optical lattices offer a unique platform for investigating disorder-driven phenomena. While static disordered site potentials have been explored in a number of experiments, a more general, dynamical control over site-energy and off-diagonal tunnelling disorder has been lacking. The use of atomic quantum states as synthetic dimensions has introduced the spectroscopic, site-resolved control necessary to engineer more tailored realisations of disorder. Here, we present explorations of dynamical and tunneling disorder in an atomic system by controlling laser-driven dynamics of atomic population in a momentum-space lattice. By applying static tunnelling phase disorder to a one-dimensional lattice, we observe ballistic quantum spreading. When the applied disorder fluctuates on time scales comparable to intersite tunnelling, we instead observe diffusive atomic transport, signalling a crossover from quantum to classical expansion dynamics. We compare these observations to the case of static site-energy disorder, where we directly observe quantum localisation.

[1] Department of Physics, University of Illinois at Urbana-Champaign, Urbana, IL 61801-3080, USA. Correspondence and requests for materials should be addressed to B.G. (email: bgadway@illinois.edu)

Over the past two decades, dilute atomic gases have become a fertile testing ground for the study of localisation phenomena in disordered quantum systems[1]. They have allowed for some of the earliest and most comprehensive studies of Anderson localisation of quantum particles[2–8], strongly interacting disordered matter[9–14] and many-body localisation[15–18]. Still, the emulation of many types of disorder relevant to real systems—e.g., crystal strain and dislocation, site vacancies, interstitial and substitutional defects, magnetic disorder and thermal phonons—will require types of control that go beyond traditional methods based on static disorder potentials[10].

The recent advent of using atomic quantum states as synthetic dimensions has broadened the cold atom toolkit with the spectroscopic, site-resolved control of field-driven transitions[19–24]. This technique has aided the study of synthetic gauge fields[19–21, 24–27], and its spatial and dynamical control offers a prime way to implement specifically tailored, dynamical realisations of disorder that would otherwise be difficult to study. However, current studies based on internal states[20, 21, 25–27], have been limited to a small number of sites along the synthetic dimension, inhibiting the study of quantum localisation in the presence of disorder.

Here, we employ our recently developed technique of momentum-space lattices[22, 28], to engineer tailored and dynamical disorder in synthetic dimensions. Our approach introduces several key advances to cold atom studies of disorder: the achievement of pure off-diagonal tunnelling disorder, the dynamical variation of disorder, and site-resolved detection of populations in a disordered system. For the case of tunnelling disorder, we examine the scenario in which only the phase of tunnelling is disordered. As expected for a one-dimensional (1D) system with only nearest-neighbour tunnelling, these random tunnelling phases are of zero consequence when applied in a static manner. When this phase disorder fluctuates on time scales comparable to intersite tunnelling, however, we observe a crossover from ballistic to diffusive transport[29]. We compare to the case of static site-energy disorder, observing Anderson localisation at the site-resolved level.

## Results

**Implementation**. Our bottom-up approach[22, 28], to Hamiltonian engineering is based on the coherent coupling of atomic momentum states to form an effective synthetic lattice of sites in momentum space (see Fig. 1). This approach may be viewed as studying transport in an artificial dimension[19], of discrete spatial eigenstates[30] (as opposed to a bounded set of atomic internal states[20, 21]) through resonant or near-resonant field-driven transitions.

Starting with $^{87}$Rb Bose-Einstein condensates of $\sim 5 \times 10^4$ atoms, we initiate dynamics between 21 discrete momentum states by applying sets of counter-propagating far-detuned laser fields (wavelength $\lambda = 1064$ nm, wavevector $k = 2\pi/\lambda$), specifically detuned to address multiple two-photon Bragg transitions, as depicted in Fig. 1a, b. Our spectrally resolved control of the individual Bragg transitions permits a local control of the system parameters, similar to that found in photonic simulators[31–35]. We can tune the strength, phase and detuning from Bragg resonance of each frequency component to control the tunnelling amplitude, tunnelling phase and site energy of each lattice link/site, respectively. This control is enabled by creating a multi-frequency beam (Laser 2 in Fig. 1a) with tailored spectral components. This is achieved by passing a single frequency laser through an acousto-optic modulator (AOM) that is driven by a tailored rf spectrum. Unique to our implementation is the direct and arbitrary control of tunnelling phases[22], and the realised tight-binding model is depicted in Fig. 1c. Here, we use this

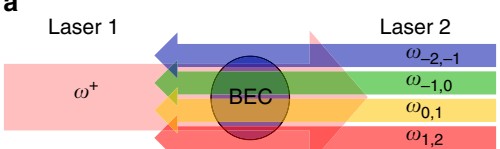

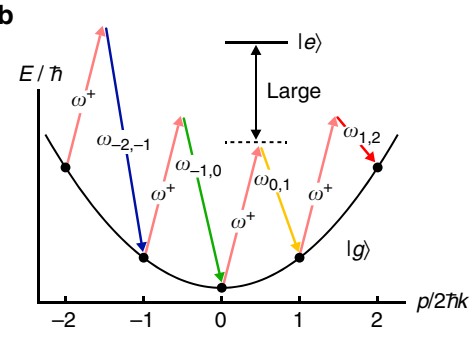

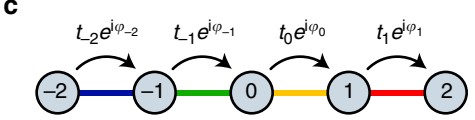

**Fig. 1** Spectroscopic control of lattice dynamics. **a** An atomic Bose-Einstein condensate (BEC) illuminated by two counter-propagating lasers, one of which (Laser 2) contains multiple discrete spectral components. **b** Energy diagram of free-particle-like momentum states coupled by counter-propagating, far-detuned Bragg laser fields (characterised by nearly identical wavevectors $k$). The spectral components $\omega_{j,j+1}$ of laser 2 are used to separately address individual Bragg transitions between momentum states $j$ and $j+1$. **c** Cartoon depiction of the effective tight-binding lattice model when all two-photon Bragg resonance conditions are matched, resulting in a flat site-energy landscape. The amplitudes and phases of the tunnelling elements $t_j e^{i\varphi_j}$ are independently controlled through the spectral components of laser 2. The lattice site energies $\varepsilon_j$ may also be independently controlled through the detunings from two-photon Bragg resonances

capability to explore the dynamics of cold atoms subject to disordered and dynamical arrangements of tunnelling elements.

Specifically, we explore disorder arising purely in the phase of nearest-neighbour tunnelling elements. In higher dimensions, such disordered tunnelling phases would give rise to random flux patterns that mimic the physics of charged particles in a random magnetic field[36–38]. In 1D, however, the absence of closed tunnelling paths renders any static arrangement of tunnelling phases inconsequential to the dynamical and equilibrium properties of the particle density. Time-varying phases, however, can have a nontrivial influence on the system's dynamical evolution.

**Diffusive transport under annealed disorder**. We engineer annealed, or dynamically varying, disorder[39–41] of the tunnelling phases and study its influence through the atoms' nonequilibrium dynamics following a tunnelling quench. Our experiments begin with all population restricted to a single momentum state (site). We suddenly turn on the Bragg laser fields, quenching on the (in general) time-dependent effective Hamiltonian

$$\hat{H}(\tau) \approx -t \sum_n \left( e^{i\varphi_n(\tau)} \hat{c}_{n+1}^\dagger \hat{c}_n + \text{h.c.} \right) + \sum_n \varepsilon_n \hat{c}_n^\dagger \hat{c}_n, \quad (1)$$

where $\tau$ is the time variable, $t$ is the (homogeneous) tunnelling energy, and $\hat{c}_n (\hat{c}_n^\dagger)$ is the annihilation (creation) operator for the

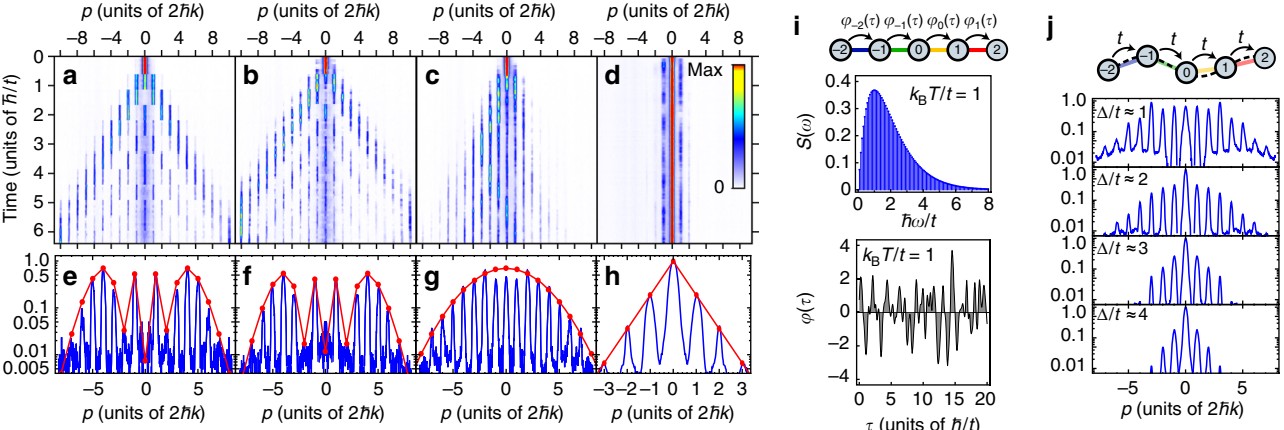

**Fig. 2** Atomic quantum walks in regular and disordered momentum-space lattices. **a–d** Nonequilibrium quantum walk dynamics of 1D atomic momentum distributions vs. evolution time for the cases of **a** uniform tunnelling, **b** random static tunnelling phases, **c** random, dynamically varying tunnelling phases characterised by an effective temperature $k_BT/t = 0.66(1)$ and **d** pseudorandom site energies for $\Delta/t = 5.9(1)$. **e–h** Integrated 1D momentum distributions (populations in arbitrary units; symmetrised about zero momentum) for the same cases as in **a–d**, after evolution times $\tau = (2.96(2)\hbar/t, 2.51(2)\hbar/t, 3.80(3)\hbar/t,$ and an average over the range $5.1(1)$ to $6.4(1)\hbar/t)$ for **e–h**. For **e**, **f**, we compare to quantum random walk distributions of the form $P_n \propto |J_n(2\tau t/\hbar)|^2$, for **g** we compare to a Gaussian distribution $P_n \propto e^{-n^2/2\sigma_n^2}$ for $\sigma_n = \sqrt{2\tau t/\hbar}$, and for **h** we compare to an exponential distribution $P_n \propto e^{-|n|/\xi}$. **i** Annealed disorder realised with tunnelling phases $\varphi(\tau)$ that vary dynamically with time $\tau$. Phases contain $N = 50$ frequency components $\omega$ that sample an ohmic spectrum $S(\omega)$, shown here peaked at effective temperature $k_BT/t = 1$. **j** Transport under pseudorandom site energies following the form $\varepsilon_n = \Delta \cos(2\pi b n + \phi)$ of an incommensurate cosine potential (*dashed line*). As in **h**, 1D momentum distributions are shown for varying pseudodisorder strengths $\Delta/t$

momentum state with index $n$ (momentum $p_n = 2n\hbar k$). The tunnelling phases $\varphi_n$ and site energies $\varepsilon_n$ are controlled through the phases and detunings of the two-photon momentum Bragg transitions, respectively. After a variable duration of laser-driven dynamics, we perform direct absorption imaging of the final distribution of momentum states, which naturally separate during 18 ms time of flight. Analysis of these distributions, including determination of site populations through a multi-Gaussian fit, is as described in ref. [22].

As a control, we first examine the case of no disorder, with all site energies set to zero and uniform, static tunnelling phases $\varphi_n(\tau) = \varphi$. Figure 2a shows the evolution of the 1D momentum distribution, obtained from time-of-flight images integrated along the axis normal to the imparted momentum, displaying ballistic expansion characteristic of a continuous-time quantum walk. For times before the atoms reflect from the open boundaries of the 21-site lattice, we find good qualitative agreement between the observed momentum distributions and the expected form $P_n = |J_n(\vartheta)|^2$, where $J_n$ is the Bessel function of order $n$ and $\vartheta = 2\tau t/\hbar$. Figure 2e shows the (symmetrised) momentum profile at time $\tau = 2.96(2)\hbar/t$ along with the Bessel function distribution for $\vartheta = 5.4$. The discrepancy between the measured evolution time and the argument of the Bessel distribution stems from the uncertainty in the measured tunnelling time $\hbar/t$, which is dependent on local laser intensity and prone to variations.

In comparison, Fig. 2b shows the case of zero site energies and static, random tunnelling phases $\varphi_n \in [0, 2\pi)$. The dynamics are nearly identical to the case of uniform tunnelling phases. This is consistent with the expectation that any pattern of static tunnelling phases in 1D is irrelevant for the dynamics of the effective tight-binding model realised by our controlled laser coupling, since these phases can be gauged away with local transformations. For this case, Fig. 2f shows the (symmetrised) momentum profile at $\tau = 2.52(2)\hbar/t$ along with the Bessel function distribution for $\vartheta = 5.35$.

While static phase disorder has little impact on the quantum random walk dynamics, we may generally expect that controlled

random phase jumps or even pseudorandom variations of the phases should inhibit coherent transport, mimicking random phase shifts induced through interaction with a thermal environment. To probe such behaviour, we implement dynamical phase disorder by composing each tunnelling phase $\varphi_n$ from a broad spectrum of oscillatory terms with randomly defined phases $\theta_{n,i}$ but well-defined frequencies $\omega_i$, the weights of which are derived from an ohmic bath distribution. Specifically, the dynamical tunnelling phases take the form

$$\varphi_n(\tau) = 4\pi \sum_{i=1}^{N} S(\omega_i)\cos(\omega_i\tau + \theta_{n,i}) / \sum_{i=1}^{N} S(\omega_i), \qquad (2)$$

where $S(\omega) = (\hbar\omega/k_BT)\exp[-(\hbar\omega/k_BT)]$, the $\theta_{n,i}$ are randomly chosen from $[0, 2\pi)$, and $T$ is an artificial temperature scale that sets the range of the frequency distribution. In this discrete formulation of $\varphi_n(\tau)$, we include $N = 50$ frequencies ranging between zero and $8k_BT/\hbar$. The frequency spectrum and dynamics for one tunnelling phase $\varphi_n(\tau)$ are shown in Fig. 2i for the case of $k_BT/t = 1$.

Figure 2c displays the population dynamics in the presence of this dynamical disorder, characterised by an effective temperature $k_BT/t = 0.66(1)$ and averaged over three independent realisations of the disorder using different phase distributions $\theta_{n,i}$. We note that the population spreads asymmetrically because we do not average over a large range of $\theta_{n,i}$ distributions. The dynamics no longer feature ballistically separating wavepackets, instead displaying a broad, slowly spreading distribution peaked near zero momentum. A clear deviation of the (symmetrised) momentum distribution from the form $P_n = |J_n(\vartheta)|^2$ describing the previous quantum walk dynamics can be seen in Fig. 2g. Instead, this more diffusive behaviour is better described by a Gaussian distribution characterised by a width $\sigma_n = \sqrt{2\tau t/\hbar}$. We find excellent agreement with a Gaussian distribution at our measured evolution time of $\tau = 3.80(3)\hbar/t$, consistent with spreading governed by an effectively classical or thermal random walk.

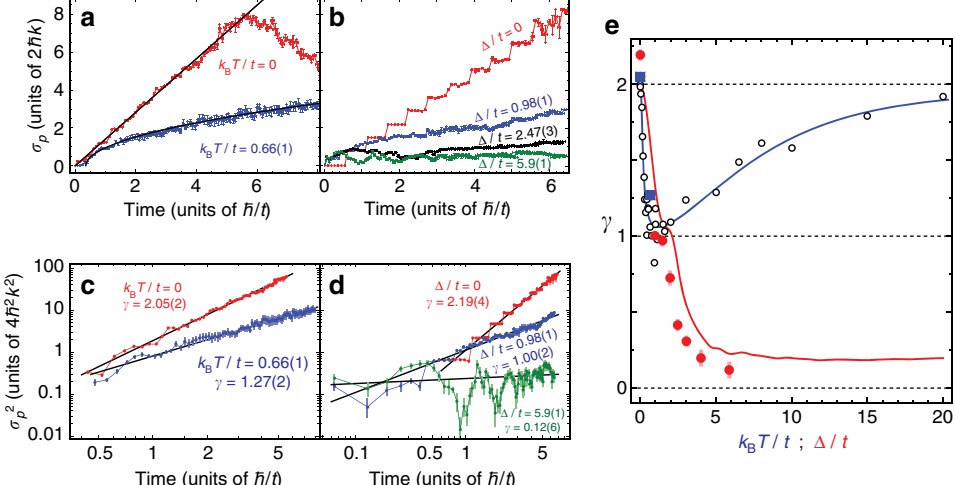

**Fig. 3** Expansion dynamics in static and dynamical disorder. **a** Momentum width $\sigma_p$ (standard deviation, units of $2\hbar k$) vs. evolution time ($\tau$, units of $\hbar/t$) for random static tunnelling phases (*red data*, labelled $k_\text{B}T/t = 0$) and random dynamical tunnelling phases (*blue data*, labelled $k_\text{B}T/t = 0.66(1)$). Overlaid as *black lines* are the predicted dynamics for ballistic ($\sigma_p = \sqrt{2}\tau$) and diffusive transport ($\sigma_p = \sqrt{2\tau}$, shifted by $0.35\hbar/t$). **b** Momentum-width dynamics for the cases of static site-energy pseudodisorder and uniform equal-phase tunnelling. The data curves relate to disorder strengths of $\Delta/t = 0$ (*red data*), $\Delta/t = 0.98(1)$ (*blue data*), $\Delta/t = 2.47(3)$ (*black data*) and $\Delta/t = 5.9(1)$ (*green data*). **c** Double logarithmic plot of the momentum variance ($\sigma_p^2$, in units of $4\hbar^2 k^2$) for the random phase data in **a**, fit to the form $V(\tau) = \alpha\tau^\gamma$. The fit-determined values of $\gamma$ are shown for each case. **d** Double logarithmic plot of the momentum variance for the static disorder data in **c**, along with power-law fits and extracted expansion exponents $\gamma$. **e** The fit-determined expansion exponents $\gamma$ plotted vs. the effective annealed disorder temperature ($k_\text{B}T/t$, *blue squares*) for dynamical disorder and vs. the disorder strength ($\Delta/t$, *red circles*) for static pseudodisorder. The *solid blue line* is a fit to numerical simulations (*open black circles*) for the case of dynamically varying phase disorder, and the *solid red line* represents numerical simulations for static pseudodisorder. All error bars denote one s.e.m

**Localisation under site-energy disorder.** Lastly, while no influence of static tunnelling phase disorder is expected in 1D, the effect of static site-energy disorder is dramatically different. Here, with homogeneous static tunnelling terms, we explore the influence of pseudorandom variations of the site energies governed by the Aubry-André model[4, 9, 12, 16]. With an irrational periodicity $b = (\sqrt{5} - 1)/2$, the site energies $\varepsilon_n = \Delta\cos(2\pi b n + \phi)$ do not repeat, and are governed by a pseudorandom distribution. For an infinite system, this Aubry-André model with diagonal disorder features a metal-insulator transition at the critical disorder strength $\Delta_c = 2t$. The expansion dynamics for the strong disorder case $\Delta/t = 5.9(1)$ are shown in Fig. 2d, with population largely restricted to the initial, central momentum order. The exponentially localised distribution of site populations (symmetrised and averaged over all profiles in the range $\tau = 5.1(1)$ to $6.4(1)\hbar/t$) is shown in Fig. 2h, along with an exponential distribution with decay length $\xi = 0.6$ lattice sites. The theoretically predicted localisation length can be described by $1/\xi = \ln(\Delta/2t)$[42], giving a value of $\xi = 0.9$ lattice sites that deviates from the value we fit from the short-time dynamics. Analogous population distributions (again symmetrised and averaged over the same time range) are shown for the cases of weaker disorder [$\Delta/t = 0.98(1), 1.96(3), 3.05(4), 4.02(9)$] in Fig. 2j. Because atoms in different lattice sites (momentum states) eventually separate spatially, we have a limited experimental timescale to observe localisation. Close to the critical point, we cannot accurately describe the population distributions with localisation lengths, though they still exhibit an apparent transition to exponential localisation for $\Delta/t \gtrsim 2$.

**Comparison of expansion dynamics.** For all of the explored cases, we study these expansion dynamics in greater detail in Fig. 3. Figure 3a examines the momentum-width ($\sigma_p$) dynamics of the atomic distributions for the cases of static and dynamic random phase disorder. For static phase disorder, we observe a roughly linear increase of $\sigma_p$ until population reflects from the

open system boundaries, while dynamical phase disorder leads to sub-ballistic expansion. In particular, for time $\tau$ measured in units of $\hbar/t$ and momentum-width $\sigma_p$ in units of the site separation $2\hbar k$, these two cases agree well with the displayed theory curves for ballistic and diffusive expansion, having the forms $\sigma_p = \sqrt{2}\tau$ and $\sigma_p = \sqrt{2\tau}$, respectively (with the latter curve shifted by $0.35\hbar/t$). To explore these two different expansions more quantitatively, we fit the momentum variance $V_p \equiv \sigma_p^2$ to a power-law $V_p(\tau) = \alpha\tau^\gamma$[43], performing a linear fit to variance dynamics on a double logarithmic scale as shown in Fig. 3c. The fit-determined expansion exponents $\gamma$ for the cases of static and dynamically disordered tunnelling phases are 2.05(2) and 1.27(2), respectively. These values are roughly consistent with a coherent, quantum random walk for the case of static tunnelling phases ($\gamma = 2$) and an incoherent, nearly diffusive random walk for the case of dynamical phase disorder ($\gamma = 1$).

The observed transport dynamics cross over from ballistic to diffusive as the effective thermal energy scale $k_\text{B}T$ approaches the coherent tunnelling energy $t$, matching our expectation that randomly varying tunnelling phases can mimic the random dephasing induced by a thermal environment. We note that similar classical random walk behaviour has been seen previously for both atoms and photons, due to irreversible decoherence[44–47] and dissipation[48, 49], and thermal excitations[50]. However, this observation is based on reversible engineered noise of a Hamiltonian parameter. These observations of a thermal random walk suggest that annealed disorder may provide a means of mimicking thermal fluctuations and studying thermodynamical properties[39] of simulated models using atomic momentum-space lattices, and by extension other nonequilibrium experimental platforms such as photonic simulators.

We also analyse the full expansion dynamics for the case of static site energy disorder in Fig. 3b, d. For homogeneous static tunnellings and thus zero disorder ($\Delta/t = 0$), we observe momentum-width dynamics similar to the case of static random tunnelling phases, but with one distinct difference: while $\sigma_p$

features a linear increase for random static phases, it increases in a step-wise fashion for uniform tunnelling phases[22]. Because our underlying implementation applies a comb of 20 discrete, equally-spaced frequency teeth to the atoms (see Fig. 1), each Bragg transition is addressed not only by an on-resonant frequency tooth, but also by 19 other frequencies in an off-resonant fashion. These off-resonant couplings add up constructively to generate jumps in the dynamics with a frequency that exactly matches the spacing between frequency teeth. By introducing random tunnelling phases onto the teeth, this constructive behaviour is suppressed, resulting in smoother dynamics. We note that the expected smooth behaviour emerges in the limit where the tunnelling is far smaller than the spacing between frequency teeth, though due to dephasing concerns we cannot work at such low tunnelling rates.

Evolution of the momentum-width ($\sigma_p$) for the site-energy disorder cases of $\Delta/t = 0.98(1)$, 2.47(3), 5.9(1) are also shown in Fig. 3b. We observe the reduction of expansion dynamics with increasing disorder, with nearly arrested dynamics in the strong disorder limit. More quantitatively, fits of the variance dynamics as shown in Fig. 3d reveal sub-ballistic, nearly diffusive expansion for intermediate disorder [$\gamma = 1.00(2)$ for $\Delta/t = 0.98(1)$], giving way to a nearly vanishing expansion exponent for strong disorder [$\gamma = 0.12(6)$ for $\Delta/t = 5.9(1)$].

The extracted expansion exponents for all of the explored cases are summarised in Fig. 3e. For static site-energy disorder (red circles), while longer expansion times than those explored ($\tau \lesssim 6.3\hbar/t$) would better distinguish insulating behaviour from sub-ballistic and sub-diffusive expansion, a clear trend towards arrested transport ($\gamma \sim 0$) is found for $\Delta/t \gg 1$. Numerical simulation (red curve) verifies this qualitative trend, but reaches a finite value of $\gamma$ due to our fits taking into account transient dynamics at short times (compared to the localisation time). The deviation from this simulation curve can possibly be attributed to the same off-resonant tunnelling terms that give rise to the step-like behaviour in Fig. 3b. Combined with the observation of exponential localisation of the site populations in Fig. 2h, j, these observations are consistent with a crossover in our 21-site system from metallic behaviour to quantum localisation for $\Delta/t \gtrsim 2$.

Our observations of a crossover from ballistic expansion ($\gamma \sim 2$) to nearly diffusive transport ($\gamma \sim 1$) for randomly fluctuating tunnelling phase disorder are also summarised in Fig. 3e. In the experimentally accessible regime of low to moderate effective thermal energies ($k_B T/t \lesssim 1$), our experimental data points (blue squares) match up well with numerical simulation (open black circles). For the magnitude of tunnelling energy used in these experiments, we are restricted from exploring higher effective temperatures ($k_B T/t \gtrsim 1$), as rapid variations of the tunnelling phases introduce spurious spectral components of the Bragg laser fields that could drive undesired transitions. Simulations in this high-temperature regime suggest that the expansion exponent should rise back up for increasing temperatures, saturating to a value $\gamma \sim 2$. This results from the fact that the time-averaged phase effectively vanishes when the time scale of pseudorandom phase variations is much shorter than the tunnelling time.

## Discussion

The demonstrated levels of local and time-dependent control over tunnelling elements and site energies in our synthetic momentum-space lattice have allowed us to perform explorations of annealed disorder in an atomic system. Such an approach based on synthetic dimensions should enable myriad future explorations of engineered Floquet dynamics[51–54] and unconventional disordered lattices[55, 56]. Furthermore, the realisation of designer disorder in a system that supports nonlinear atomic interactions[57, 58] should permit us to explore aspects of many-body localisation[59].

## Methods

**Experimental set-up.** As described in ref. [22], our experiment starts with the preparation of a $^{87}$Rb Bose-Einstein condensate containing ~$5 \times 10^4$ atoms through all-optical evaporation in a trap comprised of several optical dipole beams. The condensate is then transferred to a trap formed mainly from one of these beams (wavelength $\lambda = 1064$ nm), which we use as our lattice beam. To apply a desired Hamiltonian for the atoms and initiate dynamics, we use AOMs to imprint on the retro lattice beam multiple frequency sidebands $\omega_{j,j+1}$, which address Bragg transitions between atomic momentum states with momenta $2j\hbar k$ and $2(j + 1)\hbar k$. By addressing transitions between many adjacent momentum states, we create an effective lattice of sites in a synthetic dimension. We control the detunings from Bragg resonances as well as the amplitudes and phases of each frequency component so as to tune the site energies, tunnelling amplitudes, and tunnelling phases of each element in our lattice, respectively. We use this local parameter control to generate the many realisations of disorder presented in this work.

In this work we create 21-site lattices, but in general we can reach lattice sizes of over 50 sites. However, we cannot populate all of these sites in the experimental timeframe, due to eventual decoherence from the spatial separation of atoms in different momentum orders.

Mean field interactions in this system cause shifts in the Bragg resonance frequencies from the single-particle resonances. By directly measuring this shift to be $2\pi \times 430(40)$ Hz[58], we find a peak mean-field energy of $\mu_0 = gn_0 = \hbar \times 2\pi \times 760$ (70) Hz, relating to the peak atomic density $n_0 \approx 10^{14}$ cm$^{-3}$ at the center of our harmonic trap[60]. Here, $g = 4\pi\hbar^2 a/M_{Rb}$ for $M_{Rb}$ the mass of Rubidium and $a$ the scattering length.

**Calibrated tunnelling times.** The tunnelling times for all data were calibrated using two-site Rabi oscillations. These times are: $\hbar/t = 111.6(7)$ μs for the clean, non-disordered data (Fig. 2a, e), $\hbar/t = 115.3(9)$ μs for the random static tunnelling phases data (Fig. 2b, f), $\hbar/t = 126.4(9)$ μs for the annealed disorder data (Figs. 2c, g and 3a, c) and $\hbar/t = 158(7)$ μs averaged over all of the Aubry-André model data (Figs. 2d, h and 3b, d).

**Data availability.** All data sets presented here are available from the corresponding author upon request.

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

## Acknowledgements

We thank J. Ang'ong'a for helpful discussions and careful reading of the manuscript.

## Author contributions

F.A.A. performed the experiments with assistance from E.J.M. F.A.A. and B.G. analysed the data. All authors contributed to the preparation of the manuscript. B.G. supervised the project.

## Additional information

**Competing interests:** The authors declare no competing financial interests.

