## [Peer Review file · Nature Communications]

Reviewers' comments:

Reviewer #1 (Remarks to the Author):

The manuscript reports on the continued exploration of an experimental realization of a technique, proposed by the senior author, to use ultra-cold atoms as simulators for studying broad issues in transport phenomena. Specifically, illuminating a Bose Einstein condensate (BEC) with counter-propagating lasers, one of which is a multi-component beam, allows for the creation of a lattice in the space of recoil momentum. Stimulated two-photon Bragg transitions are then used to mimic inter-site transitions and on-site potentials that can be carefully controlled. The resulting system closely resembles a single-particle, tight-binding Hamiltonian and the laser parameters can be tuned to produce both diagonal and off-diagonal disorder, static and dynamic, with precision. The time-evolution of an initial 'site localized' state is then studied, where time-of-flight is used to separate the different momentum components in the evolved state. The range of behavior seen for various combinations of static and dynamic disorder in onsite and hopping terms is then explored, both through the direct momentum representation of evolved states as well as through the time behavior of the variance in momentum. The former is used to display characteristic shapes for diffusive, exponentially localized and ballistic motion. The latter allows for a more detailed elucidation of transport categories, such as sub-ballistic, as well as localization as a function of the ratio of characteristic scales for diagonal and off-diagonal disorder. Transitions in behavior are also explored in situations when phase dominates amplitude randomness in the off-diagonal coupling, providing a mechanism for mimicking thermal effects. Overall, I think the results are novel and of broad interest and the paper very clearly written. There are sufficient details of the basic technique to make the presentation self-contained. The potential for the method to be used in simulating other contexts, such as transport in polymer chains, in a controlled manner is also high.

I did have a couple of minor issues that I would like the authors to consider.

(i) In figures 2(b)(i) and 2(b)(ii), the observed results are contrasted with the anticipated Bessel function profile. However, in each case, the argument of the Bessel function is not what is expected from the stated value of τ . It would appear that the uncertainty is in the evolution time. If this is the case, it would be useful to state this or some indication of the reasons.

(ii) I was curious to know the limitations of the lattice size one could construct with this method. Here the multimodal beam has 20 components and the lattice is 21 sites. Given that finite size scaling is useful for addressing transport issues, any limitations on the size of the lattice that can be realized using this method would be worth specifying.

Once the authors have considered these points, I would recommend that the manuscript be accepted for publication.

Reviewer #2 (Remarks to the Author):

This manuscript reports experimental and theoretical results on the quantum simulation of transport processes in an optical lattice introducing ad-hoc decoherence processes.

The present work expands the tools available to the ultracold atom community within the quantum simulation framework and includes into the list a controlled disorder. The experimental work is of high quality, with an enormous simplicity associated to the realization and to the production of beautiful data. The novelty and quality of the realized work is certainly adequate to the level required for Nat. Commun.

However, I am not convinced that the authors have performed an optimal job in order to produce a manuscript transmitting full information to the readers. In my opinion, a published paper should represent an important occasion to transfer a full knowledge to the overall community. I am

convinced that the manuscript could be greatly improved by the authors. Notice that the authors have not used the space associated to the Supplemental Materials.

The major reserve is on the theoretical analysis of the results. The performed analysis appears qualitative, based on the power law of the momentum distribution and the strong attention on the γ parameter. I have examined the references 39-43, and even there the attention was focused on the variance of the probability distribution. On the basis of those papers, I may conclude that the analysis by the authors represents a more elaborated approach. In my opinion the authors should describe the knowledge level reached by the community, and if their theoretical analysis represents an evolution of previous work. A

The list of additional remarks is in the following.

- 1) The experimental configuration required to produce nearly 20 sidebands, controlling their spacing (not equally spaced!) is not described here. The reference to the papers previously published by the authors is not correct, because a limited number of sidebands was applied there.
 - 2) The manuscript does not report the atomic density and does not discuss if a contribution by the mean field interaction is important.
 - 3) The data in Fig.3b evidence the presence of steps in the width temporal dependence. The step explanation within the manuscript is very schematic. Instead Ref. 23 reported a similar step evolution, with a more detailed interpretation. Should that interpretation apply also to the present case? If that is the case, the authors should repeat it within the present work.
 - 4) Are all the distributions in Fig. 2d described by a localization length?
 - 5) The classical or thermal random walk associated to the iii) data of Fig. 2 is analyzed on the basis of a Gaussian distribution with w variance. The disordered configuration is analyzed on the basis of a localization length ξ . The dependence of these widths on the "conduction" parameters is not discussed within the manuscript.
 - 6) At page 4, first line, the sentence "These values are roughly consistent.." requires a reference in order to make the claimed "consistence" more quantitative.
- In conclusion the manuscript requires a revision following the lines presented above.

Ennio Arimondo

Reviewer #3 (Remarks to the Author):

This is a report on the manuscript "Ballistic, diffusive, and arrested transport in disordered momentum-space lattices" by Fangzhao Alex An, Eric J. Meier, and Bryce Gadway.

In this work, they exploit their previously established technique of creating an effective tight-binding model in momentum space by illuminating a BEC with two counter-propagating laser beams, where one beam consists of a set of discrete frequencies tailored to enable resonant transitions between different momentum classes [22]. In the present work, they specifically make use of the ability to control the phases and the offset energy of each tunnelling step individually and demonstrate their influence on transport. The paper is very well written and is rather accessible also for the non-specialist reader.

This is a great technique that enables lattice structures and control knobs complementary to those achievable by other methods. The beautiful experimental data shows not only ballistic quantum walks in these momentum-space lattices and the exponential Anderson localization in the Aubry-Andre model, but furthermore also convincingly demonstrates time-resolved control over the phase of the tunnelling elements and the resulting change in transport.

The down side of this methods lies i) in the size of the achieved lattice, which is 21 lattice sites, but might be increased further, and ii) in the absence of interaction effects, at least in the present work. Consequently, all results achieved here can be trivially simulated numerically, as they

correspond to single-particle dynamics on 21 sites. Nonetheless, as another recent manuscript by the same authors on the Arxiv shows, interactions can become relevant also in these lattices and will therefore open the route to non-trivial quantum simulations.

Therefore, I believe that the present paper is a beautiful demonstration of some novel and unique aspects of the method of momentum-space lattices and should be published.

Before publication, I would recommend that the authors consider the following points:

*Comment on Asymmetry in Fig 2,a,iii

*Caption of Fig 2b: " $>\sim 2.5$ " could mean anything from ~ 2.5 to 6. Either the authors mean just ~ 2.5 , or they should state the individual values

*Fig2c, bottom: What is the meaning of the color scale? From the caption, I deduce that it is unrelated to the colors in the topmost sketch in Fig 2c?

*Line 96: Why is dynamically varying disorder an annealed disorder? While this might be well warranted, it requires more explanation.

*Line 134ff: It might be helpful to add that these static phases can be gauged away, i.e. can be absorbed in the definition of the Wannier functions.

*Line212 and Caption Fig 3: Can you give a reason for the shift of the diffusive curves by $0.35\hbar/t$?

* Line225 and later on: Maybe talk about the effective energy scale of the noise instead of an effective thermal energy scale? The effect has nothing to do with the temperature of the atoms.

*Fig3b&d, Line 257: Cite your old work [22] also in this context; it took me some time to realize that this effect was explained there. Maybe also add numerical simulations including the off-resonant couplings, all cases should be easily comparable to numerical calculations. Furthermore, do these couplings limit the tunability of the hopping parameters?

*Fig3e: Why are there only two experimental data points for kT/t ? It would be great if one could see the upturn in γ also experimentally. There seems to be a limitation of the experimental technique that should be mentioned here.

*Fig3e: The Aubry-Andre model would predict a vanishing exponent for $\Delta/t > 2$, where the system is localized. However, as the localization length is finite, one would expect a transient dynamics during the first few tunnelling times. Is the observed dynamics consistent with this transient dynamics or does it hint of additional incoherent dynamics? Again, this can be trivially checked by numerical simulations.

Response to Reviewer 1

We thank the Reviewer for their thorough reading of our manuscript, valuable comments, and overall support of our work. We are pleased that the Reviewer has a positive view of our work and of its potential applications.

Response to comments / changes to the manuscript

Comment (#1): “In figures 2(b)(i) and 2(b)(ii), the observed results are contrasted with the anticipated Bessel function profile. However, in each case, the argument of the Bessel function is not what is expected from the stated value of τ . It would appear that the uncertainty is in the evolution time. If this is the case, it would be useful to state this or some indication of the reasons.”

Response: The reviewer is correct, and we agree that this discrepancy should be explained in the text. Our system allows for extremely precise control over the relative tunneling energies, tunneling phases, and all site energy terms. However, the magnitude of the tunneling energies (t) are dependent on the local laser intensities of two counter-propagating laser fields at the location of the atoms, and are thus prone to experimental fluctuations (i.e. due to pointing drifts, variations of the AOM diffraction efficiencies in the path of the “multi-frequency retro-reflected” beam, etc.). Thus the evolution time, when normalized to the relevant timescale of inter-site tunneling (the tunneling time, \hbar/t), has some inherent uncertainty. In figures 2(b)(i-ii), this creates a discrepancy between the measured evolution time (in terms of the real experimental duration and the calibrated tunneling rate) and the evolution time that best fits the data.

Changes: We have included a few sentences in the description of Fig. 2 (lines 137-141 in the revised manuscript) explaining how the discrepancy in tunneling time stems from an uncertainty in tunneling rates. We have also included the measured tunneling times and their uncertainties as a separate section in the methods section (lines 384-392).

Comment (#2): “I was curious to know the limitations of the lattice size one could construct with this method. Here the multimodal beam has 20 components and the lattice is 21 sites. Given that finite size scaling is useful for addressing transport issues, any limitations on the size of the lattice that can be realized using this method would be worth specifying.”

Response: One simple limitation to the number of sites that we can hope to engineer in our current setup, in terms of the Bragg frequency “teeth”, is given by the ratio of the bandwidth of our AOMs (roughly 40 MHz) to the frequency separation between each of the first-order resonances (roughly 16 kHz, set by the mass of rubidium and our chosen wavelength of 1064 nm used in a fully retro-reflected geometry). This limits us to engineering the nearest-neighbor coupling terms between roughly 2000 sites. As a relevant aside, for a fixed total laser power and beam sizes, the tunneling rate associated with the two-photon Rabi rate between neighboring momentum states would go down, due to the fact that the field strength in each frequency component of the retro-reflected path would be decreased as the number of components (roughly equal to the number of sites, N_{sites}). We note that the scaling of the two-photon Rabi rate with the intensity of the retro-reflected component is somewhat forgiving, such that (for a fixed beam size), a factor of 100 increase in the number of sites could be accommodated by only a factor of 10

increase in the laser power – that is, available laser power does not limit us from increasing to roughly 2000 lattice sites.

The real limitation that keeps us from dramatically increasing our system size is the fact that the atoms in our various synthetic lattice sites, which are actually plane-wave momentum states, eventually separate from one another. That is, the idealized experiment only happens while all of the atoms are in the “near-field,” and an effective loss of coherent dynamics sets in when the wavepackets of the different momentum orders spatially separate. This limitation is actually what motivates us to operate with such large tunneling rates (which we’ll discuss below in further responses), as we can get more tunneling events to occur before substantial loss of wavepacket overlap takes place.

For now, we are realistically limited to observing coherent transport over lattices of just a few dozen lattice sites (here we refer to the case of a fully ballistic quantum walk, where population actually spreads over all available sites). We likely could have created slightly larger lattices to avoid atoms hitting the edge of the system (i.e. the reflection seen in Fig. 3(a)), but it would not have significantly affected the transport dynamics under disorder at the timescales investigated.

Changes: We have added a few sentences describing the possible system size in the methods section (lines 371-375).

Response to Reviewer 2

We thank the Reviewer for their valuable comments and for their attention to detail. We welcome the opportunity to address the Reviewer's comments and concerns and more fully describe our work. Below we provide a point-by-point response to the Reviewer's comments, and detail related changes made to the revised manuscript.

Response to comments / changes to the manuscript

Comment (#0): "The major reserve is on the theoretical analysis of the results. The performed analysis appears qualitative, based on the power law of the momentum distribution and the strong attention on the γ parameter. I have examined the references 39-43, and even there the attention was focused on the variance of the probability distribution. On the basis of those papers, I may conclude that the analysis by the authors represents a more elaborated approach. In my opinion the authors should describe the knowledge level reached by the community, and if their theoretical analysis represents an evolution of previous work."

Response: The Reviewer is correct, that much previous work on random walks has focused on studying the variance of the spatial distribution and how this depends on, e.g., disorder strength, a controlled decoherence rate due to spontaneous emission or spatial mode mismatch, or temperature. Several of these studies have also examined the form of the spatial distribution following many "walk steps," distinguishing between ballistic expansion, classical diffusion, and exponential localization, similar to our analysis in Fig. 2.

Related to both of these points of analysis, one of the most fundamental distinctions between coherent quantum transport, incoherent classical transport, and disorder-induced localization is the scaling of the spatial variance with evolution time. In the case of quantum localization, it is expected that, following transient dynamics up until a localization time, the variance does not increase with increasing evolution time [$\sigma^2 \approx C \times \tau^0$ at long times, reaching a fixed value independent of time, τ]. For coherent, ballistic delocalization, a linear increase of the momentum width σ relates to a variance scaling of $\sigma^2 \propto \tau^2$. Lastly, incoherent, diffusive transport will lead to a scaling of $\sigma^2 \propto \tau^1$. Thus, the time exponent of the variance dynamics serves as a single quantity that clearly distinguishes between these delineating regimes of particle transport.

Our analysis in terms of this exponent, which we call γ , does not necessarily represent a huge leap or evolution in the analysis of these types of problems (at least in our opinion). It just provides a simple, single measure to succinctly describe how transport behavior changes with, for example, disorder strength or temperature. One possible issue in distilling data to this single parameter is that one needs to observe the dynamics over a long enough time period to obtain a precise fit to the slope of $\ln[\sigma^2]$ vs. $\ln[\tau]$. Even in our present study, our extracted γ values are rather imprecise, and would definitely have benefited from looking over many more tunneling times.

For reference, analyses similar to ours have been performed previously in photonic simulators, where dynamics can be tracked for long evolution times. For example, in [Naether (2013)], [Eichelkraut (2013)], and [Golshani (2014)], the slope of the variance dynamics at long times was investigated and compared directly to theory. In these studies, the extracted slopes (γ values) are quoted for particular cases, but are not concisely gathered and plotted as a function of a control parameter as in our Fig. 3(e). We now include

these references in our manuscript, and hope they can provide some relevant examples where other researchers have looked at the variance dynamics in similar details. We thank the Reviewer for pointing this issue out.

Changes: We cite additional studies in photonic systems using analyses similar to ours (line 242 and lines 258-259 in the revised manuscript).

[Naether, et al. *New J. Phys.* **15**, 013045 (2013).]

[Eichelkraut, et al. *Nat. Commun.* **4**, 2533 (2013).]

[Golshani, et al. *Phys. Rev. Lett.* **113**, 123903 (2014).]

Comment (#1): “The experimental configuration required to produce nearly 20 sidebands, controlling their spacing (not equally spaced!) is not described here. The reference to the papers previously published by the authors is not correct, because a limited number of sidebands was applied there.”

Response: In both this work and previously published work (Ref. 22), we utilize lattices with 21 sites. Fig. 1 of Reference 22 (reproduced below, on the following page) shows dynamics on lattices of 5 (subfigure d, red curve + data in e) and 21 sites (subfigure b, black curve + data in e).

We agree that Fig. 1 of this paper does not explain how to control lattice site energies, however in the caption of Fig. 1(c) and in the methods section, we explain that the detunings from Bragg resonances shift the site energies. Our previous studies of Bloch oscillations in Ref. 22 (Fig. 2) have demonstrated our system’s ability to create equal site energy shifts (relating to a linear site energy gradient) with equal frequency detunings from Bragg resonances. Creating the unequal site energy shifts for the Aubry-André disorder measurements in this work is a matter of inputting different detunings for each frequency component, and is no more experimentally challenging than our previous studies. This is because we generate this rather complex optical frequency spectrum in a rather simple way. Because the spacing between the Bragg resonances frequencies is so low, i.e. only about 16 kHz, we generate the complex rf spectrum that contains all of the important information with only a rather low-bandwidth signal generator. We then coherently map this rf spectrum onto the optical field by passing our laser through an acousto-optic modulator driven by the tailored multi-frequency rf spectrum. We agree that the text would benefit from a slight revision to include some more of these details related to the creation of disordered tight-binding models.

Previous work [22] showing generation of 5-site and 21-site lattices

Changes: We have added some text about tuning site energies and about our frequency spectrum generation in the description of our system (lines 77-85).

Comment (#2): "The manuscript does not report the atomic density and does not discuss if a contribution by the mean field interaction is important."

Response: We agree that this should be mentioned in the text. We directly measure the effect of mean-field interactions in our system by studying their shift of the Bragg resonance frequencies from the single-particle resonance. Employing a two-photon Rabi rate that is much lower than that used in our transport experiments, we determine the frequency at which we have a maximal transfer to a state with two photon momentum (we average over the transition from 0 to $2\hbar k$ and from 0 to $-2\hbar k$ to account for any small initial velocity of the atoms). We find an interaction shift of $\Delta = 2\pi \times 430$ Hz under conditions (atom number and trapping frequencies) as reported in the paper, where the trapped frequency is greatly reduced along one axis to mitigate the issue of momentum wavepacket separation. By changing the trapping frequencies to increase the atomic density, we measure shifts as high as roughly $\Delta = 2\pi \times 1200$ Hz (shown below, on the following page). Accounting for the inhomogeneous densities of our trapped condensates (as in [Stenger et al 1999]), these frequency shifts relate to peak atomic densities of $9.7 \times 10^{13} \text{ cm}^{-3}$ and $2.7 \times 10^{14} \text{ cm}^{-3}$, respectively.

(Results in prep.) Mean field interactions shift Bragg resonance away from single particle resonance

These results are partly based on our recent arXiv paper [arXiv:1702.07315], and on experimental measurements (above, preliminary) which will be incorporated into the arXiv paper.

[J. Stenger, S. Inouye, A. P. Chikkatur, D. M. Stamper-Kurn, D. E. Pritchard, and W. Ketterle, *Phys. Rev. Lett.* **82**, 4569 (1999)]

We now address the relevance of such mean-field interactions on the momentum-space dynamics of our atoms. As detailed in our recent preprint [arXiv:1702.07315], interactions effectively lead to a local attraction between bosonic atoms occupying the same momentum state (two colliding atoms in different momentum states experience a larger repulsive interaction, due to the exchange interaction). Thus, for a homogeneous mean-field energy $\mu \sim \hbar\Delta$, our typical lattice system can be modelled by a nearest-neighbor coupled many-mode nonlinear Schrodinger equation with a site-local (mostly local) attractive nonlinear energy term that scales like the fraction of atoms at that particular site times μ . In such a system, starting with all of the atoms on one interior lattice site (as performed in our experiment), one finds that the population remains self-trapped, i.e. localized due to interactions, if $\mu > 4t$. In the case of a disorder-free lattice, this dramatic effect goes away for weaker interactions. Even for $\mu \sim 3t$, there is not much qualitative difference between the quantum walk in the presence of interactions as compared to the interaction-free case. In our experiments, a characteristic tunneling energy is $t/\hbar = 2\pi \times 1273$ Hz (for a typical tunneling time of 125 microseconds). Thus, our typical tunneling energy exceeds the mean-field energy (again, relating to a roughly 400 Hz frequency shift) by about a factor of 3, and we are about a factor of 12 away from the regime where effects like self-trapping would occur.

While the present studies are restricted to the regime where interactions should not be very important, it would be exceedingly interesting in the future to explore the interplay of disorder-induced localization and nonlinear interactions (these regimes can be accessed by simply lowering our tunneling energy).

Changes: We have added a sentence on the strength of the mean field interaction in the Methods section (lines 376-383).

Comment (#3): “The data in Fig.3b evidence the presence of steps in the width temporal dependence. The step explanation within the manuscript is very schematic. Instead Ref. 23 reported a similar step evolution, with a more detailed interpretation. Should that interpretation apply also to the present case? If that is the case, the authors should repeat it within the present work.”

Response: The Reviewer is correct, and we agree that the current explanation is lacking.

Changes: We have cited our previous work, and expanded our explanation of the off-resonant step-like behavior (lines 275-289). We are able to contrast the step-like jumps in the momentum-width, which occur when all of the tunneling links are in phase, to the gradual increase that occurs for static, random tunneling phases.

Comment (#4): “Are all the distributions in Fig. 2d described by a localization length?”

Response: The $\Delta/t = 6$ distribution (not shown in Fig. 2d) is the only one that we explicitly describe in the text in terms of a fit-determined localization length of 0.6 sites. The bottom distribution of Fig. 2d ($\Delta/t = 4$) is also extremely well fit by an exponential localization with localization length 0.7 sites. The $\Delta/t = 3$ distribution deviates slightly from the expected exponential decay (as can be seen in Fig. 2(a-b)(iv)), but can also be fit and roughly described by a localization length of 1.0 sites. As can be readily seen from the other distributions of Fig. 2d, near the infinite-system transition value of $\Delta/t = 2$ and for lower disorder, the momentum profile deviates significantly from an exponential distribution (similarly for the case of $\Delta/t = 1.5$, profile not shown in Fig. 2d). While slightly above the expected transition point, the distribution for $\Delta/t = 2.5$ also has slight deviations from an exponential form, and when fit to an exponential distribution has a localization length of 1.2 sites.

For many of these cases, our determination of a localization length is impacted by the fact that we only measure the dynamics over a relatively short timescale (~ 6 tunneling times, as shown in Fig. 2a), due to the aforementioned spatial separation of momentum orders. Thus while we can observe an exponential population distribution for large disorder, the deviations are seen for moderate disorder values due to transient short-time dynamics. Moreover, the extracted magnitude of the localization length differs from the simple analytical expectation [D. J. Thouless, *Phys. Rev. B* **28**, 4272 (1983)]

$$\xi \sim 1/\ln(\Delta/2t).$$

We expect that this deviation is mainly due to transient dynamics, as the agreement between the extracted localization length and the simple expectation improves as the disorder strength is increased (for $\Delta/t = 6$ we expect a localization length of 0.9 sites, and extract from our fit an exponential decay length of 0.6 sites).

We have ongoing efforts to extend the timescales of our experiments by up to an order of magnitude, which should allow us to study localization transitions in a much more definitive manner.

Changes: We have included a sentence clarifying our usage of localization length for only large disorder values (lines 219-225).

Comment (#5): “The classical or thermal random walk associated to the iii) data of Fig. 2 is analyzed on the basis of a Gaussian distribution with w variance. The disordered configuration is analyzed on the basis of a localization length ξ . The dependence of these widths on the “conduction” parameters is not discussed within the manuscript.”

Response: We agree, and we now define these parameters in terms of the system’s basic conduction parameters. For example, as described in the response to the last comment, the exponential decay length ξ (ξ) can be related to the localization length. In particular, in mapping the Aubry-André model of a quasidisordered potential onto the physics of a 2D tight-binding electron in a transverse magnetic field (Harper-Hofstadter model), ξ specifies the localization length along one spatial direction, with the electron delocalized in the other direction [D. J. Thouless, *Phys. Rev. B* **28**, 4272 (1983)]. In this picture, the localization length ξ is related to the basic conduction parameters Δ (the disorder parameter in the Aubry-André model) and t (the tunneling energy) as $\xi^{-1} \sim \ln(\Delta/2t)$, diverging at the transition point.

Similarly, the Gaussian width (formerly w , now defined as σ_n) of the thermal random walk can be related to the tunneling energy t . Specifically, for a given evolution time τ , we expect a Gaussian width of $\sigma_n = \sqrt{2\tau t/\hbar}$.

Changes: We have removed the “ w ” parameter describing Fig. 2(b,iii) in favor of “sigma” to match Fig. 3. We now define both sigma and xi in terms of the basic conduction parameters of our model system (Caption of Fig. 2 and main text lines 187-191 and 212-216).

Comment (#6): “At page 4, first line, the sentence “These values are roughly consistent..” requires a reference in order to make the claimed “consistence” more quantitative.”

Response: We agree, and have made the appropriate additions.

Changes: We have added the expected values of gamma for the ballistic and diffusive situations in this sentence, so that the claimed consistence is quantitatively substantiated (lines 248 and 250).

Response to Reviewer 3

We thank the Reviewer for their thorough reading of our text and for their detailed comments. We welcome the opportunity to address the Reviewer's comments and concerns and more fully describe our work. Below we provide a point-by-point response to the Reviewer's comments, and detail related changes made to the revised manuscript.

Response to comments / changes to the manuscript

Comment (#1): "Comment on Asymmetry in Fig 2,a,iii"

Response: The form of the annealed disorder takes in random phases $\theta_{n,i}$ dependent on the site index n and the frequency component index i . When averaged over many such combinations of random $\theta_{n,i}$ values, the averaged dynamics should look perfectly symmetric and diffusive. However, the data in Fig. 2(a,iii) are an average over only three $\theta_{n,i}$ values (with several identical of the experiment at each time and for each $\theta_{n,i}$ value), and thus results in asymmetric spreading.

Changes: We have added some explanation of the asymmetry in the text, and an explicit statement that three sets of phase values $\theta_{n,i}$ were used (lines 178-181).

Comment (#2): "Caption of Fig 2b: " $>\sim 2.5$ " could mean anything from ~ 2.5 to 6. Either the authors mean just ~ 2.5 , or they should state the individual values"

Response: While we state the individual values in the main text, we agree that the figure caption also needs this information. We have also decided to quote exact evolution times instead of using \sim , both in the caption and in the main text.

Changes: Caption of Fig. 2b and description of Fig 2b (lines 135, 151, 191, and 210) now quote exact evolution times.

Comment (#3): "Fig2c, bottom: What is the meaning of the color scale? From the caption, I deduce that it is unrelated to the colors in the topmost sketch in Fig 2c?"

Response: The colors in Fig. 2c were intended to correspond monotonically with the magnitude of the phase plotted, unrelated to the colors in the upper sketch. We have decided that this color scale is redundant with the curve, and have changed the plot to have a uniform gray filling. We thank the Reviewer for bringing this to our attention.

Changes: The rainbow colors in Fig.2c, bottom have been replaced with a uniform gray filling.

Comment (#4): “Line 96: Why is dynamically varying disorder an annealed disorder? While this might be well warranted, it requires more explanation.”

Response: “Annealed disorder” is terminology defined to mean time-varying disorder, in contrast with “quenched disorder” which refers to static disorder. This wording has been used in theoretical proposals in the context of cold atoms [Osterloh (2005)] (theoretical, because such dynamical disorder does not show up naturally in optical or magnetic potentials), but has previously been used extensively to discuss time-varying randomness in condensed matter systems – such as in classic [Rapaport (1972)] and more recent [Belitz (2000)] treatments of transport in dynamical disorder.

We agree that our usage of the term should be better defined or motivated, and have included these citations.

[K. Osterloh, M. Baig, L. Santos, P. Zoller, and M. Lewenstein, *Phys. Rev. Lett.* **95**, 010403 (2005).]

[D. C. Rapaport, *J. Phys. C: Solid State Phys.* **5**, 2813 (1972).]

[D. Belitz, T. R. Kickpatrick, and T. Vojta, *Phys. Rev. Lett.* **84**, 5176 (2000).]

Changes: The references [Osterloh (2005)], [Rapaport (1972)], and [Belitz (2000)] are now cited when mentioning “annealed disorder” (line 104).

Comment (#5): “Line 134ff: It might be helpful to add that these static phases can be gauged away, i.e. can be absorbed in the definition of the Wannier functions.”

Response: We agree.

Changes: We include a sentence on gauging away the static phases (lines 149-150).

Comment (#6): “Line212 and Caption Fig 3: Can you give a reason for the shift of the diffusive curves by 0.35hbar/t?”

Response: Unfortunately, we do not have a good explanation. We believe it may be due to the same off-resonant effects that cause the step-like behavior in the clean lattice (Fig. 3(b), red line), but have not determined the exact cause of the shift.

Changes: N/A

Comment (#7): “Line225 and later on: Maybe talk about the effective energy scale of the noise instead of an effective thermal energy scale? The effect has nothing to do with the temperature of the atoms.”

Response: While we can see the possible confusion with the actual temperature of the atoms, we prefer to keep the current terminology of effective thermal energy. We feel that this description serves the intended role in helping to visualize the form of the disorder as the analog of a thermal bath (Fig. 2(c)). In our system, the actual temperature of the atoms, so long as they are sufficiently cold, plays no direct role in their dynamical behavior in momentum space. However, actual noise on the field-driven transitions, in addition to “engineered noise” as we explore, can play the role of an effective thermal bath that leads to relaxation of the momentum-space transport. We would thus expect the dynamics to “thermalize” with this noise, i.e. in the case of equal site-energies that atoms would roughly populate all states (in a finite-sized lattice) equally.

This picture of an effective temperature scale is also important for understanding what role dynamical parameter variations can play when considering not non-equilibrium quench dynamics, but rather initializing a ground state and then coupling to a “thermal bath.” This could allow us to probe the energy spectrum of our engineered system in a simple way, gaining access to rich information not easily accessible by other means. Using the language of an effective temperature makes a direct connection to more traditional kinds of cold atom experiments.

Lastly, similar language of an effective temperature scale is often used when discussing driven transitions between other types of discrete single-particle quantum states – namely spin states of a single particle. This parallel is most relevant when considering particles with a spin degree of freedom that are essentially divorced from real sources of thermal relaxation (think of very dilute gases in vacuum). Starting from a ground spin state, interactions with a coherent drive field can result in “negative” temperatures, i.e. through population inversion, and the decoherence of the spin due to a noisy drive field can be interpreted as it thermalizing with a bath.

Changes: N/A

Comment (#8): “Fig3b&d, Line 257: Cite your old work [22] also in this context; it took me some time to realize that this effect was explained there. Maybe also add numerical simulations including the off-resonant couplings, all cases should be easily comparable to numerical calculations. Furthermore, do these couplings limit the tunability of the hopping parameters?”

Response: We agree with the need for a citation – we overlooked this! In responding to Reviewer #2, we felt that the explanation presented in this text is not sufficient, and have added more details.

We feel that adding numerical simulations to these plots, which already contain 3-4 datasets, would clutter the presentation in Fig. 3b, d. Further, in previous work [22] we have already presented exact numerical simulations of these “steps”.

And finally, yes, the off-resonant effects impose an upper limit on the range of the hopping parameters t : when t/h is on the order of the spacing between adjacent frequency components (~ 16 kHz), we begin to unintentionally excite the wrong momentum states – i.e. we lose spectral resolution. For the studies presented here, we work in a regime where t is much smaller than this spacing ($t \sim 1$ kHz), so that the off-

resonant effects manifest only in the step-like behavior, which does not qualitatively alter the dynamical evolution.

Changes: The explanation of the step-like behavior now includes more detail and cites our old work [22] (lines 275-289).

Comment (#9): “Fig3e: Why are there only two experimental data points for kT/t ? It would be great if one could see the upturn in γ also experimentally. There seems to be a limitation of the experimental technique that should be mentioned here.”

Response: We could have taken more experimental data for relatively low values of kT/t ($kT/t < 1$), but unfortunately, the more interesting large kT/t regime is experimentally unviable. The Reviewer is correct that there is an experimental limitation (at least for the employed tunneling energies t) that keeps us from accessing the limit $kT > t$. Basically, the temporal variations in the tunneling phases can be alternatively be understood as leading to fluctuations in the frequency components (for example, a linear phase “ramp” is equivalent to a static frequency shift). When the rate at which the phase variations occur are large enough, they can lead to shifts of the individual frequency teeth of such a large magnitude that they actually begin to address neighboring resonances. For the tunneling energies employed (roughly $t/\hbar = 2\pi \times 1250$ Hz), this effect becomes problematic when kT begins to exceed t (note that, as specified in the text, the actual spectrum of applied discrete phase modulation frequencies ω extends all the way up to $8kT/\hbar$).

This limitation is explained in the last paragraph of the results section (lines 286-291, or lines 324-329 in the revised manuscript): the phases of the annealed disorder begin to vary too rapidly, driving undesired transitions.

In the future, the “high-temperature” regime could be accessed if we were to operate with lower tunneling energies, such that higher ratios of kT/t could be accessed without driving off-resonant excitations.

Changes: N/A

Comment (#10): “Fig3e: The Aubry-Andre model would predict a vanishing exponent for $\Delta/t > 2$, where the system is localized. However, as the localization length is finite, one would expect a transient dynamics during the first few tunneling times. Is the observed dynamics consistent with this transient dynamics or does it hint of additional incoherent dynamics? Again, this can be trivially checked by numerical simulations.”

Response: The observed dynamics are consistent with transient dynamics. Because our momentum orders eventually spatially separate, we can’t go to a long enough evolution time to observe a sharp distinction between delocalization and localization setting in after a finite localization time. However, we can see true qualitative differences between transport for $\Delta < 2t$ and transport at very high disorder amplitudes Δ . We have simulated this behavior and found good qualitative agreement with our slope (γ) data in Fig. 3e, shown below (on the following page). The solid red line is new, derived from

fitting the variance dynamics of the numerical simulations over the same timescale as the experimental data.

Updated Fig. 3e, with (red) simulation curve of Aubry-Andre disorder

The simulation curve does not reach zero for large disorder because we fit the full dynamics, including the transient behavior (same range of fitting times for both theory and experiment). The differences between the data and the simulation curve can possibly be attributed to a systematic shift in the measured tunneling rate (due to fluctuations in lattice laser intensities, as discussed earlier), or possibly also to effects of laser phase noise, the separation of momentum wavepackets, or even the nonlinear interactions. The agreement (or disagreement) between actual time traces of the data and theory is shown more explicitly for two of the larger disorder values ($\Delta/t = 2.5$ and 6) in the plot below.

Dynamics for $\Delta/t = 2.5$ (blue) and 6 (green) with solid simulation curves

Changes: Fig. 3e now includes a simulation curve (red) for the Aubry-Andre disorder data. Discussion of this additional curve has also been added to the caption and main text (lines 306-312).

Reviewer #1 (Remarks to the Author):

The authors have made significant changes to the manuscript in response to the comments of all three referees. The revisions made certainly address the issues I raised and the other additions have also made the presentation more self-contained. I am entirely comfortable recommending publication of the revised manuscript.

REVIEWERS' COMMENTS:

Reviewer #2 (Remarks to the Author):

The authors have taken into account all the remarks of my previous report. The manuscript can be accepted for publication in the present form.

Reviewer #3 (Remarks to the Author):

The authors have adequately addressed all points raised in my previous report. The paper appears to be ready for publication and should be accepted.

Reviewer 1 (Remarks to the Author)

The authors have made significant changes to the manuscript in response to the comments of all three referees. The revisions made certainly address the issues I raised and the other additions have also made the presentation more self-contained. I am entirely comfortable recommending publication of the revised manuscript.

Response to Reviewer 1

We thank the Reviewer for their comments and support of our work.

Reviewer 2 (Remarks to the Author)

The authors have taken into account all the remarks of my previous report. The manuscript can be accepted for publication in the present form.

Response to Reviewer 2

We thank the Reviewer for their comments and support of our work.

Reviewer 3 (Remarks to the Author)

The authors have adequately addressed all points raised in my previous report. The paper appears to be ready for publication and should be accepted.

Response to Reviewer 3

We thank the Reviewer for their comments and support of our work.